# Diagnostic Value of Single-Photon Emission Computed Tomography/Computed Tomography Scans with Tc-99m HDP in Cervicogenic Headache

**DOI:** 10.3390/jcm9020399

**Published:** 2020-02-02

**Authors:** Pyung Goo Cho, Tae Woo Kim, Gyu Yeul Ji, Sang Hyuk Park, Mi Jin Yun, Dong Ah Shin

**Affiliations:** 1Department of Neurosurgery, Ajou University College of Medicine, Suwon-si 16499, Korea; nsdrcpg@ajou.ac.kr; 2Department of Neurosurgery, Yonsei University College of Medicine, Seoul 03722, Korea; TWKIM416@yuhs.ac; 3Department of Neurosurgery, Cham Teun Teun research institute, Seoul 06072, Korea; jivago91@gmail.com; 4Department of Neurosurgery, Hwalgichan Hospital, Ilsan-si 10500, Korea; carespine@naver.com; 5Department of Nuclear Medicine, Yonsei University College of Medicine, Seoul 03722, Korea; YUNMIJIN@yuhs.ac

**Keywords:** cervicogenic headache, single-photon emission computed tomography/computed tomography, computed tomography, spinal injection, diagnosis

## Abstract

A cervicogenic headache (CEH) is difficult to diagnose due to its varied pathology. We evaluated the usefulness of single-photon emission computed tomography/computed tomography (SPECT/CT) in diagnosing CEH and its interventional treatment. Retrospectively, 23 patients diagnosed with CEH between March 2016 to August 2018 were allocated to SPECT/CT (*n* = 11) and control (*n* = 12) groups. The SPECT/CT group was further stratified into SPECT/CT(+) and SPECT/CT(-) groups according to the presence of positive findings. Patients in the SPECT/CT group underwent an intra-articular injection at a radiologically verified lesion site, whereas those in the control group underwent third occipital nerve block. Clinical outcomes were evaluated with the visual analog scale (VAS), neck disability index (NDI), and global perceived effect (GPE) scale at baseline, and at one, three, and six months postoperatively. The SPECT/CT group showed less VAS, NDI, and GPE scores at six months postoperatively (2.91 ± 2.30 vs. 4.42 ± 1.62, *p* = 0.08; 38.00 ± 16.54 vs. 48.7 ± 12.40, *p* = 0.093; 2.00 ± 1.41 vs. 3.17 ± 1.11, *p* = 0.039). Successful responders at six months postoperatively were higher in the SPECT/CT(+) group than in the control group (75% vs. 0%). SPECT/CT can identify arthritic changes and accurately define therapeutic targets.

## 1. Introduction

A cervicogenic headache (CEH) is defined as a secondary headache in which the pain is perceived in the head or face, and the associated nociceptors are located in the neck [1]. The pathogenetic mechanism of CEH involves convergence between the upper cervical and trigeminal afferents in the trigeminocervical nucleus [2,3,4]. Pathognomonic features of CEH include unilateral pain radiating from the neck to head, a temporal pain pattern, and pain induced by improper motion and external pressure on the neck [5,6]. However, the clinical features of CEH overlap with those of a tension-type headache or a migraine [5,7,8,9,10]. Moreover, even healthy individuals may experience CEH [11]. The clinical criteria for the diagnosis of CEH remain undefined [4], and the identified radiographic findings are not pathognomonic of CEH [12]. Moreover, discrepancy exists between the outcomes of clinical diagnosis and objective testing for CEH [4]. Therefore, physicians remain skeptical about CEH, which eventually hinders its diagnosis and treatment [4]. Therefore, a novel method for diagnosing CEH is needed urgently.

Single-photon emission computed tomography/computed tomography (SPECT/CT) is increasingly used to diagnose facet joint arthritis based on both the morphology and physiology of the facet joint [13]. Russo et al. showed that SPECT/CT facilitated improved treatment outcomes for back pain by allowing precise localization of the arthritic facet joint [13]. Dolan et al. also reported that SPECT(+) patients reported significantly greater improvement after treatment of facet joint arthritis as compared to SPECT(-) patients [14]. These studies show that SPECT/CT can identify the pain generators in the lumbar spine. Accordingly, we hypothesized that SPECT/CT could also visualize these pain generators in CEH. Results from previous studies have shown that noxious stimulation of the atlanto-occipital, lateral atlanto-axial (AAJ) [15], and C2–3 zygapophyseal joints [16] induce CEH. Demonstration of the pain-generating lesions by SPECT/CT would enable the objective diagnosis of CEH and the accurate treatment of pain-generating lesion, thereby improving the treatment results.

To the best of our knowledge, no previous study has investigated SPECT/CT imaging findings in patients with CEH. Therefore, in this study, we evaluated the role of SPECT/CT with Tc-99m hydroxydiphosphonate (HDP) for diagnosing CEH and its interventional treatment.

## 2. Materials and Methods

This is a retrospective, observational study. The study was conducted in accordance with the Declaration of Helsinki, and the protocol was approved by the Ethics Committee of Severance Hospital, Yonsei University College of Medicine, which was obtained before study initiation (Approval number: 4-2019-0484).

### 2.1. Study Overview

This study is a retrospective analysis of patients with CEH who had been recruited at the time before and after the introduction of SPECT/CT in our institution. (Figure 1) After SPECT/CT was installed in our institution, all patients with clinically suspected CEH had a SPECT/CT unless they had a contraindication. Nine men and 14 women were included in this study. All variables were investigated by examination of patient medical records, a private surgical database, and telephonic interviews. Because of the retrospective study design, the requirement for informed consent of the patients was waived. The diagnosis of CEH was made by one neurosurgeon (DAS) with more than 10 years of experience in spinal interventions. All spinal injections were administered by one neurosurgeon (DAS) using the same interventional protocol. Data were collected and analyzed by two neurosurgeons (PGC and TWK). A follow-up evaluation of the clinical outcome was conducted at one month, three months, and six months postoperatively. The six month follow-up data were available for all patients and were analyzed to assess the role of SPECT/CT in diagnosing CEH.

### 2.2. Patient Selection

Between March 2016 and August 2018, a consecutive series of 23 patients who were diagnosed with CEH and received spinal injection at our hospital were identified. The age of the patients at the time of receiving spinal injection ranged from 29 to 87 years (mean age ± standard deviation, 58.4 ± 14.86 years). The inclusion criteria were the following: (1) Precipitation of pain in the neck and occipital region on neck movement and/or sustained awkward head positioning, (2) presence of confirmatory evidence of CEH by diagnostic anesthetic blockade, and (3) unilaterality of the headache, without shifting sides [5]. Of the 69 patients who received a spinal injection in the upper cervical at our hospital, 46 were not included to the study because they did not meet the inclusion criteria or were subject to exclusion criteria. Patients whose treatment options have changed were excluded from the study, as opposed to the treatment methods in this study. The exclusion criteria were as follows: Presence of (1) bilateral headache, (2) dissecting aneurysm, (3) meningitis, (4) neck-tongue syndrome, (5) C2 neuralgia, (6) greater occipital neuralgia, (7) Barré-Lieou syndrome, (8) malignancy, (9) infectious disease, or (10) a history of surgery [5]. The patients were divided into two groups-the SPECT/CT (*n* = 11; patients diagnosed with CEH who underwent SPECT/CT) and control (*n* = 12; patients with CEH who did not undergo SPECT/CT) groups. Patients in the SPECT/CT group were further classified into SPECT/CT(+) and SPECT/CT(-) groups according to the presence or absence of positive findings. The control group was composed of patients who did not take SPECT/CT before the installation of SPECT/CT and patients who refused to take SPECT/CT due to contraindication.

### 2.3. Radiographic Evaluation

Cervical radiography (X-ray), CT, and magnetic resonance images were obtained for all patients before they received the spinal injections. The cervical facet joints were examined using the modified Pathria grade by axial magnetic resonance imaging (MRI) [17]. The degree of degeneration on the radiographic and CT images were scored from 0 to 5 (0 = normal; 1 = hypertrophy; 2 = erosion; 3 = instability; 4 = sclerosis; and 5 = narrowing of the facet joint). The degree of degeneration observed on MRI was scored from 0 to 3 (0 = normal; 1 = joint space narrowing (mild); 2 = joint space narrowing and sclerosis or hypertrophy of the facet joint (moderate); 3 = severe osteoarthrosis with narrowing and sclerosis of the facet joint and formation of osteophytes (severe)) [17].

### 2.4. SPECT/CT

Patients were injected intravenously with 1110 mBq ± 10% of Tc-99m hydroxydiphosphonate (Tc-99m HDP, Richmond, CA, USA). Planar whole body and spot images were obtained 3.5–4 h after Tc-99m HDP injection. Additionally, SPECT imaging of the C-spine was performed using the Symbia T16 SPECT/CT system (Siemens Medical Solutions Inc., Ann Arbor, MI, USA) and reconstructed into axial, sagittal, and coronal planes. CT images were obtained at 2 mm slice thickness, 110 kV voltage, and the minimum current was set in real-time using the Care Dose 4D method, with a 512 × 512 matrix using a standard filter. SPECT images of the cervical spine were acquired with 1.0X zoom and a 256 × 256 matrix with a step-and-shoot scan mode (32 frameset, 30-s/frame, and 18 angles). SPECT image reconstruction was performed with CT-based attenuation correction and a Hanning three-dimensional (3D) post-filter (cut-off frequency, 0.85 cycles/cm), with a 3D Flash iterative reconstruction (eight patients and 16 iterations). The SPECT acquisition took approximately 16 min, whereas the CT acquisition took no more than 10 s. The reconstructed, attenuation-corrected SPECT data were co-registered with CT and viewed in multiplanar projections. All SPECT/CT scans were reviewed on a dedicated Syngo workstation (Siemens, VA60A software, Erlangen, Germany) by two nuclear medicine physicians with more than 10 years of experience, and differences in opinion were resolved by discussion to reach a consensus. A positive finding (SPECT/CT(+)) was defined as the observation of higher relative activity than uptake around a normal facet joint in the cervical spine on SPECT/CT (Figure 2A–C).

### 2.5. Injection Technique

The spinal injection method was chosen after considering the patient’s symptoms and examining the radiographic, CT, and MRI findings (Table 1). In the SPECT/CT group, an intra-articular injection at the AAJ and a cervical-third- (C3) occipital nerve (C-TON) injection were performed according to the lesion confirmed by the SPECT/CT. In the control group, only the C-TON injection was performed.

### 2.6. C-TON Block

For the C-TON block, patients were placed in the prone position with a pillow under the chest to allow for slight neck flexion. The C-arm (Philips Healthcare, Best, Netherlands) was placed over the head and neck in an anteroposterior direction. Under fluoroscopic guidance, the C-arm was rotated in a cephalad-caudad direction until the lateral C2–3 joint was well visualized. Only one needle was inserted, and it was moved to each of the target points sequentially. The insertion site was determined by placing the needle on the skin of the patient’s neck over the middle target region. The TON passed somewhat horizontally across the joint at a level that ranged between the top and just below the bottom of the intervertebral foramen [18]. To ensure the TON blockade was within this range, 0.3 mL of local anesthetic was injected at each of the three target points that lay vertically over the middle of the joint. The target points were a high point at the level of the apex of the C3 superior articular process, a low point at the level of the bottom of the C2–3 intervertebral foramen, and a middle point halfway between the other two.

The operator inserted the needle and directed it to each of the three target points sequentially. At each target, the position of the needle tip was checked and recorded on a lateral fluoroscopic image, and 0.3 mL of 0.75% ropivacaine with steroids was injected.

### 2.7. Intra-Articular Injection at the AAJ

The patients were placed in the prone position, and the C-arm was placed over the head and neck in an anteroposterior direction. The C-arm was rotated in a cephalad-caudad direction until the lateral AAJ was clearly visualized. The needle insertion site was marked on the skin overlying the lateral third of the AAJ. The skin was prepped and draped, maintaining sterility, and a skin wheal was raised with local anesthetic injection at the insertion site. Thereafter, a 22-G blunt needle was advanced toward the posterolateral aspect of the inferior margin of the inferior articular process of the atlas (C1). This approach prevented contact with the C2 nerve root and dorsal root ganglion (DRG), which crossed the posterior neck and touched the bone to establish the correct depth. At this point, a lateral view was obtained. The needle was withdrawn slightly, directed toward the posterolateral aspect of the lateral AAJ, and advanced by a few millimeters. Usually a distinctive pop is felt when the needle enters the joint cavity. (Figure 3A,B) Care was taken to avoid the vertebral artery that lay lateral to the lateral AAJ as it coursed between the C1 and C2 foramina. After careful negative aspiration for blood or cerebrospinal fluid, 0.1–0.2 mL of water-soluble nonionic contrast agent was injected with real-time fluoroscopy to rule out accidental intravascular injections and to confirm the intra-articular insertion of the needle tip.

Anteroposterior and lateral views were obtained to ensure the retention of the contrast agent to the joint cavity without diffusion to the surrounding structures, specifically the epidural space or posteriorly to the C2 DRG, as this adversely affects the specificity of the block. The anteroposterior view usually demonstrates the bilateral concavity of the joint with the retained contrast agent inside the joint space, and rarely shows the lateral AAJ space communicated by the median and contralateral AAJ. After careful negative aspiration, 1.0 mL of 0.75% ropivacaine with steroids was injected.

### 2.8. Assessment of Clinical Outcome

In our outpatient’s clinic, outcome assessments were conducted at baseline, and at one, three, and six months after the procedure by a nurse who was specialized in pain management. The participants’ neck or occipital pain intensity was assessed using an 11-point subjective visual analog scale (VAS; 0 = no pain, 10 = worst pain imaginable). The effect of neck pain on their quality of life was assessed using the neck disability index (NDI), with scores ranging from 0–50. These results were expressed on a scale ranging from 0% (no disability) to 100% (maximum disability). The global perceived effect (GPE) according to the 7-point Likert scale (1 = significant improvement, 7 = significant deterioration) was also used to assess patient satisfaction and symptom improvement [19]. Adverse events during the treatment and follow-up were individually recorded.

The outcome measure was the number of successful responders to treatment at each follow-up time-point. A successful response was determined based on the methodology described previously, with some modifications [20,21]. The strict criteria of successful response were defined as all of the followings were satisfied, and if any of them was satisfied, we defined it as the loose criteria: >50% (or 4-point) reduction from baseline in the neck and occipital VAS score, ≥30% decrease from baseline in the NDI, and ≤2 points on the GPE scale.

Patients were prescribed nonsteroidal anti-inflammatory drugs postoperatively. For the first month after the procedure, the patients were instructed to avoid changing their previously prescribed medications. The prescribed doses of each analgesic, except for opioids, were increased or decreased based on the remnant pain intensity of the patient evaluated at each follow-up visit. Patients who required alterations in the dosage of analgesic medication or who wanted alternative treatments were considered as treatment failures and were excluded from the study after that follow-up visit.

### 2.9. Statistical Analyses

Statistical analyses were performed using SAS version 9.4. software (SAS Ins., Cary, NC, USA). All values of continuous variables were expressed as mean ± standard deviation and *p*-values were calculated by independent *t*-tests. All values of categorical variables were expressed as count (%) and *p*-values were calculated by chi-square tests or Fisher’s exact tests. A *p*-value < 0.05 was considered to be statistically significant.

## 3. Results

Twenty-three patients with CEH were included in this study (Table 1). Among them, 11 patients underwent SPECT/CT scans. The mean age of patients in the SPECT/CT group was higher than that in the non-SPECT/CT group (65.4 ± 12.4 years vs. 52.1 ± 14.5 years; *p* = 0.029). There was no statistical difference in sex ratio, location of pain, and duration of symptoms between the two groups (*p* = 0.680, *p* = 0.400, and *p* = 0.485, respectively). Only patients with unilateral lesions were included in the study according to the previous study’s diagnostic criteria and this study’s inclusion criteria. In addition, no bilateral hot uptake was observed in this study. In the SPECT/CT(+) group, five patients underwent intra-articular injections and three patients underwent TON block. Patients in both the SPECT/CT(-) and control groups underwent TON block. CEH associated with lower cervical lesions were not observed in this study.

The values of clinical outcome variables, such as VAS, NDI, and GPE, for each group are shown in Table 2, Table 3 and Table 4. Both the SPECT/CT and control groups showed postoperative improvement at all follow-up visits. The SPECT/CT group showed less VAS, NDI, and GPE scores at six months postoperatively (2.91 ± 2.30 vs. 4.42 ± 1.62, *p* = 0.08; 38.00 ± 16.54 vs. 48.7 ± 12.4, *p* = 0.093; 2.00 ± 1.41 vs. 3.17 ± 1.11, *p* = 0.039, respectively). Individual VAS, NDI, and GPE scores at six months postoperatively were higher in the SPECT/CT(+) group than in the control group (87.5% vs. 50.0%, 75.0% vs. 33.3%, and 87.5% vs. 58.3%, respectively) (Table 3). The percentages of successful responders in the SPECT/CT(+) group at one, three, and six months postoperatively were 50.0%, 37.5%, and 75%, respectively. The percentage of patients who exhibited ≥50% reduction in VAS scores, ≥30% reduction in NDI scores, and ≤2 GPE at six months postoperatively were higher in the SPECT/CT(+) group than in the control group (75% vs. 0%) (Table 4). There was a significant difference in the number of successful responders between the SPECT/CT(+) and SPECT/CT(-) groups (*n* = 6 vs. 0, *p* = 0.061) (Table 4). However, by applying the loose criteria of successful response, the statistical difference had disappeared (Appendix A). There was no difference in the radiography findings and Pathria grade between the success and non-success responders (Table 5).

In the SPECT/CT group, eight patients (73%) showed positive findings. Among the patients in the SPECT/CT(+) group, six (75%; *p* = 0.059) had abnormal radiography findings but only three patients (38%) showed abnormal CT findings. Cohen’s kappa was calculated and showed agreement between the SPECT/CT and radiography findings; however, agreement between the SPECT/CT and CT findings was low (κ = 0.377 and 0.247, respectively). All patients in the SPECT/CT(+) group had abnormal MRI findings. In this group, a good correlation was noted between the SPECT/CT and MRI findings of the facet joint. The SPECT / CT (+) group tended to show abnormalities on X-ray and MRI, but there was no statistical difference (Table 6).

## 4. Discussion

To date, CEH is diagnosed based on the clinical features and the findings of physical examinations, as a diagnosis of CEH is confirmed by positive results from a diagnostic anesthetic blockade [22]. However, these methods are not always accurate [23]. The symptoms of CEH overlap with those of a tension-type headache and a migraine [5,7,8,9,10]. Although MRI allows the identification of disc protrusion, dorsal ligament thickening, dural compression with spinal stenosis, and nerve compression, and can confirm these changes in the facet joint capsule, these findings are not pathognomonic of CEH. Therefore, the diagnostic criteria for CEH are not clinically practical and have poor inter-rater reliability and specificity [23], and so a superior method for diagnosing CEH is urgently required. In this study, we retrospectively evaluated the role of SPECT/CT for diagnosing patients with CEH. We found that patients who had positive SPECT/CT findings demonstrated a better clinical outcome after receiving a spinal injection. The VAS and NDI scores of these SPECT/CT(+) patients improved by 88% and 75%, respectively. In contrast, SPECT/CT(−) patients showed poor outcomes.

Although CEH has variable etiologies, arthritic changes of the upper cervical facet joint (C2/3) is the main cause of CEH [24]. Because SPECT/CT can visualize the pathologic area by measuring osteoblastic activity, it is easier to identify the pain generator [13]. Before SPECT/CT was introduced, a blind TON block was the most commonly administered treatment for CEH [25]. The introduction of SPECT/CT enabled the administration of specific lesion-based injections. Functional changes could be confirmed with earlier versions of SPECT; however, locating the exact lesion site was difficult due to low spatial resolution. Advancements in SPECT/CT, however, have enabled accurate identification of the lesion by merging the benefits of SPECT with CT. Patients with symptomatic facet joint arthritis are more likely to present positive findings on SPECT/CT. Dolan et al. reported that patients who underwent facet joint injections were more likely to show positive findings on SPECT/CT than those who did not show any positive findings on SPECT/CT at one and three months postoperatively [14]. Ackerman et al. reported that pain-relief and reduction of disability were more common in SPECT/CT(+) patients than in the SPECT/CT(−) patients [26]. These results indicate that SPECT/CT can identify symptomatic facet joint arthritis.

In this study, the SPECT/CT findings and clinical results were mismatched in 18% of the patients, and treatment failure was observed in 25% of the SPECT/CT(+) patients.

There are several reasons for this mismatch. SPECT/CT can show changes in the bone but not in the soft tissue. Thus, SPECT/CT cannot identify the causes of CEH that involve soft tissue lesions, such as neural inflammation, nerve entrapment, or soft tissue tumor. Another reason for mismatch is that SPECT/CT shows bone remodeling, thus, even elevated non-pathological bone activity may result in a false-positive finding. Furthermore, despite significant degeneration in the bone, SPECT/CT finding may be negative if the bone remodeling has already been completed and if the stabilization stage has concluded. According to Russo et al., 5.5% of the patients with poor facet joint arthritis had positive findings on SPECT/CT, whereas 31% of the patients with severe facet joint arthritis had negative findings on SPECT/CT [13]. Cases of arthritis that are not particularly severe cannot be identified by SPECT/CT. For these reasons, some patients with negative SPECT/CT findings also demonstrate favorable outcomes.

The present study had a few limitations. First, some patients with CEH may theoretically appear negative on SPECT/CT; for instance, patients with inflammation of the greater or lesser occipital nerve. However, the number of such cases is expected to be small. Second, patients in the present study only had upper cervical lesions, but some previous studies have reported that lower cervical lesions can also cause CEH [27,28]. In future studies, we will investigate the relationship between the pathophysiology of CEH and the lower cervical spine using SPECT/CT. Third, this report comprised a small sample size. The sample size was small because only patients confirmed with the diagnostic block were included. Fourth, only one physician participated in the clinical diagnosis, so measurement bias may exist. Fifth, among the various treatments of CEH, only spinal injection was studied in our research. In the near future, large-scale multi-center studies will address these problem.

## 5. Conclusions

The present study suggests that SPECT/CT is a useful diagnostic tool for CEH. SPECT/CT clearly demonstrated arthritic changes in the cervical spine and accurately defined therapeutic targets. Patients with SPECT/CT(+) benefited markedly from spinal injections. Thus, this imaging approach supported the diagnosis of CEH and enhanced the therapeutic outcome of spinal injections.

## Figures and Tables

**Figure 1 jcm-09-00399-f001:**
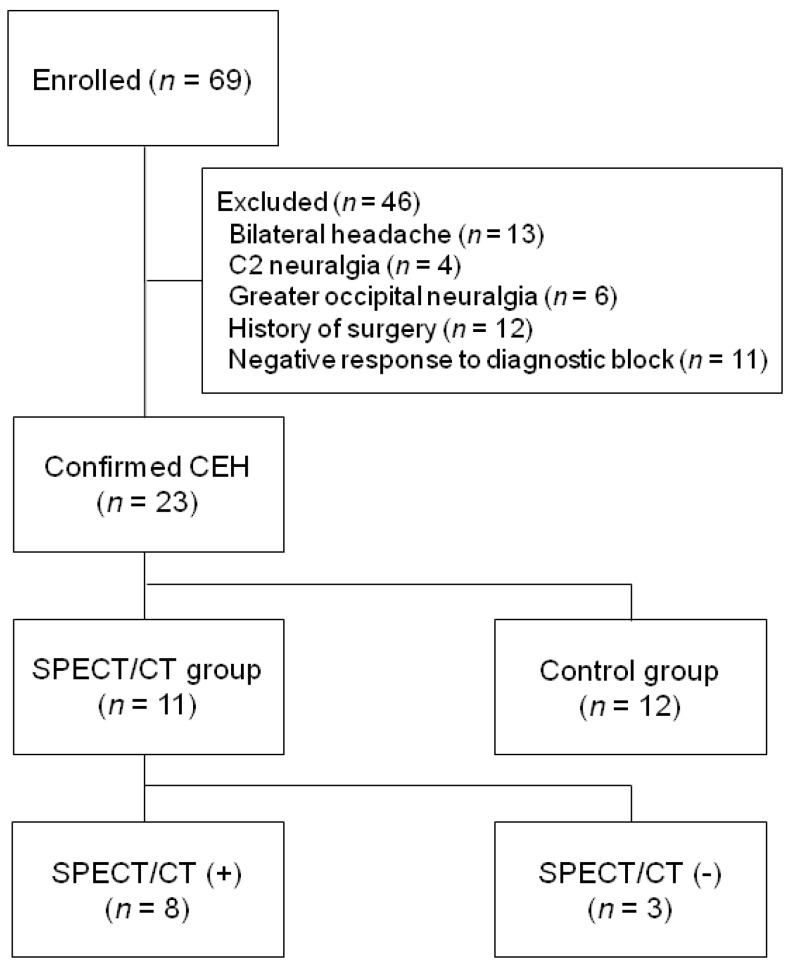
Study flow diagram.

**Figure 2 jcm-09-00399-f002:**
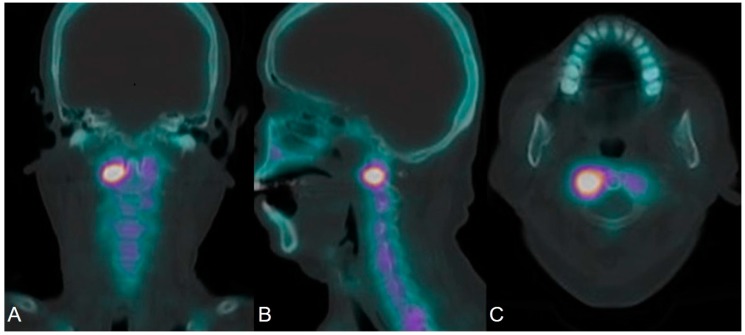
Single-photon emission computed tomography/computed tomography (SPECT/CT) with Tc-99m hydroxydiphosphonate (HDP) of a patient with cervicogenic headache shows high-grade uptake at the right lateral atlantoaxial joint. (**A**) coronal section; (**B**) sagittal section; (**C**) axial section. SPECT/CT, single-photon emission computed tomography/computed tomography.

**Figure 3 jcm-09-00399-f003:**
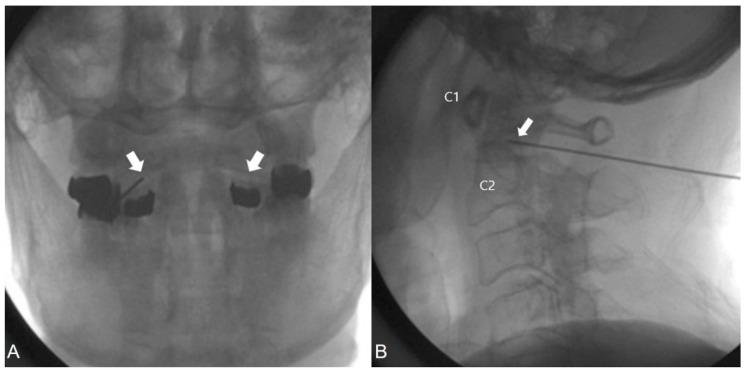
Anterior-posterior and lateral fluoroscopy of the cervical spine with a needle inserted into right lateral atlantoaixal joint. White arrow, intra-articular joint space; (**A**) anteroposterior view; (**B**) lateral view.

**Table 1 jcm-09-00399-t001:** Patient demographics.

	SPECT/CT (+) Group	SPECT/CT (-) Group	Control Group	*p*
Sex (male : female)	4:4	1:2	4:8	0.765
Age (mean ± SD)	71.3 ± 7.9	49.67 ± 6.66	52.1 ± 14.5	0.333
Pain location				
Neck + occipital	5	2	6	0.400
Occipital	7	3	12	0.217
Postauricular	1	0	0	0.478
Duration of symptom, months	5.5 ± 1.1	5.0 ± 1.2	4.8 ± 2.3	0.485
Intervention types				0.008
Third occipital nerve block	3	3	12	0.014
Intraarticular injections				0.359
C1-2	5	0	0	
C2-3	2	3	0	
C3-4	1	0	0	

SPECT/CT, single-photon emission computed tomography/computed tomography; SD, standard deviation.

**Table 2 jcm-09-00399-t002:** Treatment outcomes in the SPECT/CT and control groups.

	SPECT/CT Group	Control Group	*p*
VAS (mean, SD)			
Baseline (a)	7.18 ± 1.08	6.58 ± 0.90	0.162
1 month after treatment	2.91 ±1.58	4.25 ±1.76	0.069
3 months after treatment	3.09 ± 1.76	3.50 ± 1.09	0.505
6 months after treatment	2.91 ± 2.30	4.42 ± 1.62	0.082
NDI (mean %, SD)			
Baseline (a)	60.55 ± 7.65	60.00 ± 6.15	0.852
1 month after treatment	40.18 ± 14.30	47.17 ± 11.77	0.213
3 months after treatment	38.55 ± 16.20	46.50 ± 11.19	0.182
6 months after treatment	38.00 ± 16.54	48.67 ± 12.40	0.093
GPE (mean, SD)			
1 month after treatment	2.18 ± 0.87	2.75 ± 1.22	0.215
3 months after treatment	2.09 ± 1.14	2.92 ± 0.79	0.055
6 months after treatment	2.00 ± 1.41	3.17 ± 1.11	0.039

VAS, visual analog scale; SPECT/CT, single-photon emission computed tomography/computed tomography; SD, standard deviation; NDI, neck disability index; GPE, global perceived effect.

**Table 3 jcm-09-00399-t003:** Observed number of patients who satisfied the individual parameters (VAS, NDI, and GPE) at each follow-up.

Parameters	SPECT/CT (+) Group		SPECT/CT (-) Group		Control Group	
	Follow-up (Months)	Number (%) *N* = 8	Follow-up (Months)	Number (%) *N* = 3	Follow-up (Months)	Number (%) *N* = 12
≥50% (or ≥ 4-point) Reduction in VAS	1	6 (75%)	1	2 (66.7%)	1	7 (58.3%)
	3	6 (75%)	3	0 (0%)	3	6 (50%)
	6	7 (87.5%)	6	0 (0%)	6	6 (50%)
≥30% Reduction in NDI	1	4 (50%)	1	0 (0%)	1	5 (41.7%)
	3	3 (37.5%)	3	0 (0%)	3	5 (41.7%)
	6	6 (75%)	6	0 (0%)	6	4 (33.3%)
≤2 point in GPE	1	5 (62.5%)	1	1 (33.3%)	1	7 (58.3%)
	3	6 (75%)	3	0 (0%)	3	2 (16.7%)
	6	7 (87.5%)	6	0 (0%)	6	3 (58.3%)

SPECT/CT, single-photon emission computed tomography/computed tomography; VAS, visual analog scale; NDI, neck disability index; GPE, global perceived effect.

**Table 4 jcm-09-00399-t004:** The observed number of patients who meet the strict criteria successful response at six months follow-up between each group.

	Success (*n* = 6)	Non-Success (*n* = 17)	*p*-Value
SPECT/CT group	6 (100.00)	5 (29.41)	0.005
Control group	0 (0.00)	12 (70.59)	
SPECT/CT (+) group	6 (100.00)	2 (40.00)	0.061
SPECT/CT (-) group	0 (0.00)	3 (60.00)	
SPECT/CT (+) group	6 (100.00)	2 (14.29)	0.001
Control group	0 (0.00)	12 (85.71)	

The strict criteria of successful response were defined as all of the followings were satisfied: >50% (or 4-point) reduction from baseline neck and occipital VAS, ≥30% decrease from baseline NDI, and ≤2 points on the GPE scale. SPECT/CT, single-photon emission computed tomography/computed tomography; VAS, visual analog scale; NDI, neck disability index; GPE, global perceived effect.

**Table 5 jcm-09-00399-t005:** Comparison of radiography findings and Pathria grades between the patient groups with favorable and unfavorable outcomes at six months follow-up postoperatively.

	Success (*n* = 6)	Non-Success (*n* = 17)	*p*-Value
Radiography findings			0.613
Normal	1 (20.00)	7 (41.18)	
Abnormal	4 (80.00)	10 (58.82)	
Hypertrophy	0 (0.00)	1 (10.00)	
Erosion	1 (25.00)	5 (50.00)	
Instability	1 (25.00)	1 (10.00)	
Sclerosis	0 (0.00)	0 (0.00)	
Narrowing	2 (50.00)	2 (20.00)	
Pathria grade (MRI)			0.115
Normal	0 (0.00)	6 (42.86)	
Abnormal	6 (100.00)	8 (57.14)	
Mild	1 (16.67)	4 (50.00)	
Moderate	2 (33.33)	4 (50.00)	
Severe	3 (50.00)	0 (0.00)	

Pathria grade: Normal, normal joint space; mild, mild joint space narrowing; moderate, moderate joint space narrowing with sclerosis or hypertrophy; severe, severe joint space narrowing with sclerosis or hypertrophy; MRI, magnetic resonance imaging.

**Table 6 jcm-09-00399-t006:** Comparison of radiography findings and Pathria grades between the SPECT/CT(+)and SPECT/CT(-) groups.

	SPECT/CT(+) (*n* = 8) (%)	SPECT/CT(-) (*n* = 3) (%)	*p*
Radiography findings			0.183
Normal	2 (25.00)	2 (66.67)	
Abnormal	6 (75.00)	1 (33.33)	
*Hypertrophy*	1 (16.67)	0 (0)	
*Erosion*	1 (16.67)	1 (100)	
*Instability*	2 (33.33)	0	
*Sclerosis*	0 (0)	0	
*Narrowing*	2 (33.33)	0	
Pathria grade (MRI)			0.273
Normal	0 (0)	1 (33.33)	
Abnormal	8 (100)	2 (66.67)	
*Mild*	1 (12.50)	2 (100)	
*Moderate*	4 (50.00)	0 (0)	
*Severe*	3 (37.50)	0 (0)	

Pathria grade: Normal, normal joint space; mild, mild joint space narrowing; moderate, moderate joint space narrowing with sclerosis or hypertrophy; severe, severe joint space narrowing with sclerosis or hypertrophy. SPECT/CT, single-photon emission computed tomography/computed tomography; MRI, magnetic resonance imaging.

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
