# Peer review of "Diagnostic Value of Single-Photon Emission Computed Tomography/Computed Tomography Scans with Tc-99m HDP in Cervicogenic Headache"

_jcm, 2020, doi:10.3390/jcm9020399_

Round 1

Reviewer 1 Report

This study evaluated the relationship between SPECT/CT with Tc-99m HDP and treatment results in patients with CEH.

Major points

Generally, uptake in SPECT/CT with Tc-99m HDP is frequently observed (e.g., elder patients with cancer frequently have abnormal uptakes in facet joints which are not related with cancer). In addition, some of these uptakes can be changed over time. I am not sure whether uptake in SPECT/CT with Tc-99m HDP at only one time point is related with CEH or not.

In the paper, the following three groups were defined:

1 CEH(+) SPECT/CT with positive findings,

2 CEH(+) SPECT/CT without positive findings,

3 CEH(+) no SPECT.

The selection criteria of 1+2 and 3 (the way to determine who receives SPECT/CT) is unclear for me.

Intro. “The clinical criteria  for  the  diagnosis  of  CEH  remain  undefined  [4]” and  M&M. “Between March 2016 and August 2018, 23 patients who were diagnosed with CEH” If clinical criteria of CEH is undefined, diagnosis of CEH should be carefully performed. Although inclusion and exclusion criteria are described in the paper, these criteria were confirmed by only one doctor or by several doctors?

In this retrospective study, were the 23 patients selected consecutively or not?

“A positive finding (SPECT/CT(+)) was defined as the observation of higher relative activity than uptake around a normal facet joint in the cervical spine on SPECT/CT.” How do the authors deal with unilateral and bilateral uptakes? If “(3) unilaterality of the headache” is included in the criteria, bilateral uptake might not be related with CEH.

“A positive finding (SPECT/CT(+)) was defined as the observation of higher relative activity than uptake around a normal facet joint in the cervical spine on SPECT/CT.” “Second, patients in the present study only had upper cervical lesions, but some previous studies have reported that lower cervical lesions can also cause CEH” How do the authors deal with uptakes in facet joints of lower and higher neck? Uptakes in facet joints of lower neck might not be related with CEH.

“The spinal injection method was chosen after considering the patient's symptoms and examining the  radiographic,  CT,  and  MRI  findings.” Please clarify that SPECT/CT was used for determining spinal injection method or not.

“Outcome assessments were conducted at baseline, and at 1, 3, and 6 months after the procedure by a nurse who was specialized in pain management and was blinded to the patient’s treatment.” This study is retrospective. If so, is it possible for the nurse to perform the outcome assessment without knowing the patient’s treatment? It is quite unnatural for clinical situation.

“Patients who required alterations in the dosage of analgesic medication or who wanted alternative treatments were considered as treatment failures and were excluded from the study after that follow-up visit.” This procedure may bias the results of this paper.

“A  successful  response  was  defined  as  the  following:  >50%  (or  4-point) reduction from baseline in the neck and occipital VAS score, ≥30% decrease from baseline in the NDI, and ≤ 2 points on the GPE scale.” In addition to this definition, I recommend to use loose and strict definition and to create tables which correspond to Table 4. Because VAS, NDI, and GPE are based on patients’ subjective assessment, the definition of successful response is difficult to define.

Minor points

Tables 1 should be further divided into the results of CEH(+) SPECT/CT with positive findings and those of CEH(+) SPECT/CT without positive findings.

In the intro. and title, please clarify that SPECT/CT is performed with Tc-99m HDP.

Please clarify the experience of two experienced nuclear medicine physicians.

Please add Tc-99m HDP to the caption of Figure 1.

Please add “VAS, visual analog scale; GPE, global perceived effect” to footnote of Table 2.

“however, locating the exact lesion site was difficult because of the diffuse image quality.” “the diffuse image quality” is not adequate. I recommend to use low spatial resolution.

“All  patients  in  the  SPECT/CT(+)  group  had abnormal MRI findings. In this group, a good correlation was noted between the SPECT/CT and MRI findings of the facet joint (TABLE 5).” Table 5 does not show the correlation between the SPECT/CT and MRI findings (Table 5 shows correlation between success/failure response and MRI findings).

“However, there was no correlation between the Pathria grade on MRI and radiography findings (TABLE 6).” In table 6, indirect comparison between Pathria grade on MRI and radiography findings is shown.

Author Response

December 18, 2019

Journal of Clinical Medicine

Dear Reviewer

Thank you very much for your letter on December 12, 2019 with regard to our manuscript (jcm-666100) together with the comments from the four reviewers. I am sending herewith our revised manuscript.

I   believe the manuscript has improved satisfactorily and hope it will be accepted for publication in Journal of clinical medicine.

Sincerely yours,

Pyung Goo Cho/ Dong Ah Shin

Professor.

Reviewer 2 Report

This paper describes a clinical study using SPECT/CT for CEH diagnosis. 23 patients were involved in the retrospective study and split into SPECT/CT and control group. The clinical outcomes were evaluated using three subjective scores. The results showed statistically significant difference between SPECT/CT(+) group and the control group, indicating the feasibility of using SPECT/CT for therapeutic target localization in CEH patients. However, the sample size in this study may not be sufficient. Considering the sample size for SPECT/CT (-) group is only 3, any statistically analysis with this group would be unreliable. Besides, some other issues need to be addressed, including:

Line 46: “… a novel method to for diagnosing …” Remove “to” Line 55: “We hypothesized that SPECT/CT could also visualize these pain generators in the cervical spine.” This hypothesis seems has already been studied by the following published paper

Matar, H.E.; Navalkissoor, S.; Berovic, M.; Shetty, R.; Garlick, N., Casey, A.T.; Quigley, A.M. Is hybrid imaging (SPECT/CT) a useful adjunct in the management of suspected facet joints arthropathy?. International orthopaedics, 2013, 37, 865-870.

Please modify the current hypothesis.

Line 78: As authors want to claim a new diagnosis modality for CEH patients, please provide the criteria for the previous CEH diagnosed patients. Line 80: please add the standard deviation to the age Line 139, 146, 160: please use 0.3 mL instead of .3 mL Figure 2 caption: please be consistent and use “intra-articular” Line 185: Please give one or two references to these successful responder criteria. Giving a small sample size in this study, any subjective scores may easily be manipulated to show statistically significant results. Line 191: please give the number of patients that are excluded from the study. The sample size for SPECT/CT(-) group is relatively too small for reliable statistical analyses. Please address any problem this may bring to the results and show any correction factor/method used to ameliorate this extreme small sample size effect.

Author Response

(The authors gave the same response as above.)

Round 2

Reviewer 1 Report

Thank you for revising the manuscript.

Major points

1

“In the paper, the following three groups were defined: 1 CEH(+) SPECT/CT with positive findings, 2 CEH(+) SPECT/CT without positive findings, 3 CEH(+) no SPECT. The selection criteria of 1+2 and 3 (the way to determine who receives SPECT/CT) is unclear for me.

The purpose of this study was to investigate the efficacy of SPECT/CT in CEH, which has been shown to be useful in facet joint arthritis. Therefore, we divided into groups using SPECT/CT and groups not using SPECT/CT. To examine the false positives and false negatives of SPECT/CT, the groups using SPECT/CT were divided into groups with positive findings and those without. The findings were summarized in the tables and further described in the result section.”

In my opinion, this response is not sufficient for me. You only show the reason and results of grouping. Please clarify the criteria to determine who received SPECT/CT.

2

“The diagnosis of CEH was made by one neurosurgeon (DAS) with more than 10 years of experience in spinal interventions.”

Because the diagnosis was made by only one doctor, please clarify this as limitation.

3

“Only patients with unilateral lesions were included in the study according to the previous study's diagnostic criteria and this study's inclusion criteria. In addition, no bilateral hot uptake was observed in this study.”

These results should be added to the results section.

4

“However, no CEH associated with lower cervical lesions was observed in this study.”

These results also should be added to the results section.

5

“In our hospital, pain-specialized nurses are measuring patient outcome in almost every patients. In order to maintain the objectivity of measurement, the patients’ treatments are blinded when measuring patient outcome. The pain-specialized nurses measured the patient's outcome data without knowing about the patient's treatment and recorded it on the electronic charts. And we collected and analyzed the data from the electronic charts.”

For daily clinical practice, I think that this procedure is unnatural. I cannot believe this statement. 

6

“In order to improve the quality of the study and make the groups more purely, patients with altered treatment options were excluded from the study.”

Please clarify this as limitation.

7

“The table using the loose definition is presented belows. In the analysis that defined the loose definition of successful treatment, significant difference between the SPECT/CT and control groups disappeared. We are concerned that attaching this table can confuse the interpretation of the results.”

Please add this table of loose definition to Supplement.

8

Because of 1 and 6 of my comments, the results of this paper may be strongly biased.

Minor points

9

“oflow  spatial  resolution.”

Spell miss.

Author Response

Dear reviewer.

Thank you for inviting us to submit a revised draft of our manuscript entitled, “Diagnostic value of single-photon emission computed tomography/computed tomography scans with Tc-99m HDP in cervicogenic headache” to Journal of Clinical Medicine. We also appreciate the time and effort you and each of the reviewers have dedicated to providing insightful feedback on ways to strengthen our paper. Thus, it is with great pleasure that we resubmit our article for further consideration. We have incorporated changes that reflect the detailed suggestions you have graciously provided. We also hope that our edits and the responses we provide below satisfactorily address all the issues and concerns you and the reviewers have noted.

To facilitate your review of our revisions, the following is a point-by-point response to the questions and comments delivered in your letter dated Dec 25, 2019.

Again, thank you for giving us the opportunity to strengthen our manuscript with your valuable comments and queries. We have worked hard to incorporate your feedback and hope that these revisions persuade you to accept our submission.

Sincerely,

Dong Ah Shin

Professor

Department of Neurosurgery

Yonsei University College of Medicine
